# The Effectiveness and Safety of Multi-Strain Probiotic Preparation in Patients with Diarrhea-Predominant Irritable Bowel Syndrome: A Randomized Controlled Study

**DOI:** 10.3390/nu13030756

**Published:** 2021-02-26

**Authors:** Barbara Skrzydło-Radomańska, Beata Prozorow-Król, Halina Cichoż-Lach, Emilia Majsiak, Joanna Beata Bierła, Ewelina Kanarek, Agnieszka Sowińska, Bożena Cukrowska

**Affiliations:** 1Department of Gastroenterology, Medical University of Lublin, Jaczewskiego 8, 20-950 Lublin, Poland; barbara.radomanska@gmail.com (B.S.-R.); prozorow1@wp.pl (B.P.-K.); lach.halina@wp.pl (H.C.-L.); 2Faculty of Medicine, Cardinal Stefan Wyszynski University, Wóycickiego 1/3, 01-938 Warsaw, Poland; e.majsiak@interia.pl; 3Department of Pathomorphology, The Children Memorial Health Institute, Aleja Dzieci Polskich 20, 04-730 Warsaw, Poland; j.bierla@ipczd.pl (J.B.B.); e.kanarek@ipczd.pl (E.K.); a.sowinska@ipczd.pl (A.S.)

**Keywords:** irritable bowel syndrome, probiotics, *Lactobacillus*, *Bifidobacterium*, IBS-SSS, IBS-GIS

## Abstract

The aim of this randomized double-blind placebo-controlled study was to evaluate the effectiveness and safety of multi-strain probiotic in adults with diarrhea-predominant irritable bowel syndrome (IBS-D). The patients were randomized to receive a mixture of *Lactobacillus*, *Bifidobacterium*, and *Streptococcus thermophilus* strains or placebo for eight weeks. Primary endpoints included changes in symptom severity and improvement assessed with the IBS Severity Scoring System (IBS-SSS) and Global Improvement Scale (IBS-GIS). The probiotic in comparison with placebo significantly improved the IBS symptom severity (the change of total IBS-SSS score from baseline −165.8 ± 78.9 in the probiotic group and −105.6 ± 60.2 in the placebo group, *p* = 0.005) and in the specific scores related to the severity of pain (*p* = 0.015) and the quality of life (*p* = 0.016) after eight weeks of intervention. The probiotic group indicated an improvement in symptoms with the use of the IBS-GIS compared with the placebo group after four (*p* = 0.04) and eight weeks (*p* = 0.003). The occurrence of adverse events did not differ between study groups. In conclusion, the multi-strain probiotic intervention resulted in a significant improvement in IBS symptoms evaluated with the use of both IBS-SSS and IBS-GIS scales. The results suggest that the studied probiotic preparation is well tolerated and safe and can offer benefits for patients with IBS-D. (registration number in Clinicaltrials.gov NCT 04662957).

## 1. Introduction

Irritable bowel syndrome (IBS) is one of the most common functional gastrointestinal diseases and is characterized by recurrent abdominal pain associated with changes in stool frequency and form, with no recognized underlying pathological or organic etiology [1]. The incidence of IBS varies significantly between countries and regions [2]. A meta-analysis published in 2012 revealed that a pooled estimate of international IBS prevalence was 11.2% [3]. As there are no specific biomarkers for IBS and the disease is diagnosed clinically based on the Rome criteria, epidemiological data may depend on the criteria used [4]. A recent epidemiological survey using the Rome IV criteria revealed that the prevalence of IBS in the general population globally is 4.1% in comparison with 10.1% when the Rome III criteria were used [5]. Depending on the predominant alteration in bowel habits, IBS can be classified into diarrhea-predominant IBS (IBS-D), constipation-predominant IBS, and IBS with mixed bowel habits, and IBS-D accounts for about one-third of all IBS cases but with more severe outcomes [2].

The pathogenesis of IBS is multifactorial with no single etiology but with wide spectrum of abnormalities including motility disturbances, visceral sensitivity, altered mucosal immune functions, increased intestinal permeability, disturbances in gut microbiota composition, or altered sensory neuron activation and central nervous system processing [1,6]. There has been increasing focus on the possible connections between these conditions [6,7,8]. It is currently assumed that the intestinal dysbiosis may be a main factor responsible for most of the pathological conditions described [9,10]. The intestinal dysbiosis, also known as the gut microbiota dysbiosis, is defined as a condition in which there is an imbalance of the microorganisms and their functional activities within the intestines [11].

The gut-associated microbiota is considered a “super organ” that performs many functions in the human body: it affects the digestion and absorption of nutrients; it is responsible for the development and functioning of immunity, especially non-specific immunity, and the induction of anti-inflammatory responses; it shapes the intestinal epithelial barrier; and it affects the microbiota–gut–brain axis [12]. It should be emphasized that the functioning of the microbiota–gut–brain axis is based on two-way interdependent communication. On the one hand, dysbiosis can induce changes in well-being, abdominal pain, and motility disorders; on the other hand, stress can affect not only the degree of pain sensation, but also the composition and functional activity of the intestinal microbiota [13].

Several studies have reported significant alterations in the gut microbiota that may promote the development and persistence of IBS [14,15,16,17]. A recently published meta-analysis of the molecular signature of intestinal microbiota showed a significantly lower abundancy of *Lactobacillus*, *Bifidobacterium,* and *Faecalibacterium prausnitzi*, but not the *Bacteroides-Prevotella* group, *Escherichia coli*, or other species in IBS patients compared with healthy controls [18]. Subgroup analysis demonstrated that a downregulation in the colonization of *Lactobacillus* and *Bifidobacterium* species was mainly observed in IBS-D patients, which suggests that the presence of these bacteria could be responsible for intestinal homeostasis in IBS-D.

Thus, the modification of the microbiota composition begins to play an increasingly important role in the treatment of IBS [19]. In this respect, the supplementation of IBS patients with probiotics seems very promising. Probiotics are defined as live microorganisms which, when administered at the correct dose, provide a significant health benefit to the host [20]. They can improve gut microbiota dysbiosis and limit pathogenic bacterial colonization [21]. In addition, some probiotics induce an anti-inflammatory response, while others can modulate visceral hypersensitivity or significantly improve the integrity of the intestinal epithelium and decrease the gut barrier permeability [22,23,24]. Probiotics have been reported to be effective in reducing global and specific IBS symptoms (such as abdominal pain or flatulence), as well as in improving patients’ quality of life [25,26,27,28,29,30]. Although experts agree that probiotics have beneficial effects in IBS and are safe for patients, they fail to agree on the composition, dosage, and duration of probiotic intervention. A recently published meta-analysis of thirty-five randomized clinical trials showed that supplementation with multi-strain probiotics can be superior in comparison with single-strain preparations in IBS symptom improvement [28]. However, experts still believe that further research into the role of probiotics is required if probiotic preparations are to be adopted as a treatment for IBS.

Therefore, the aim of the current randomized, double-blind, placebo-controlled (RDBPC) study is to evaluate the effectiveness and safety of a multi-strain probiotic preparation containing *Lactobacillus, Bifidobacterium*, and *Streptococcus thermophilus* species in IBS-D patients.

## 2. Subjects and Methods

### 2.1. The Study Design

This was a RDBPC parallel-group study carried out at outpatient clinics between November 2019 and May 2020. The study was conducted in accordance with the ethical principles set out in the Declaration of Helsinki Guideline on Good Clinical Practice. The trial was approved by institutional Bioethical Committee of the Children’s Memorial Health Institute and was given decision number 6/KBE/2018. All the patients were informed about the aim, study design, and protocol, and those who agreed to participate were asked to sign the informed consent form prior to study inclusion. The study was registered at ClinicalTrials.gov 10 December 2020 and received the trial number NCT 04662957.

### 2.2. Subjects

The study included female and male patients aged 18–70 years diagnosed with IBS-D according to the Rome III criteria [31], which means they passed at least one stool assessed in Bristol Stool Form Scale as type 5, 6, or 7 for at least 2 days a week [32]. IBS severity was evaluated using the IBS Severity Scoring System (IBS-SSS) [33], and patients with at least a moderate type of IBS (IBS-SSS score >175) were included.

The exclusion criteria were coexisting severe diseases such as malignancies, uncontrolled hypertension, uncontrolled diabetes mellitus, serious neurological disorders, psychosis, respiratory disorders (asthma, chronic obstructive pulmonary disease), hyper- or hypothyroidism; hepatic, renal, or cardiac dysfunction; chronic bowel disorders other than IBS such as inflammatory bowel diseases, celiac disease, gastroenteritis, gastric and duodenal ulcers, associated constipation; parasitic or bacterial intestinal infestation/infections; diagnosed lactose intolerance; pregnancy or lactation. Patients who were treated with antibiotics (including rifaximine) and/or supplemented with probiotics during the 12 weeks (3 months) prior to study entry were also excluded from the study. Other exclusion criteria included the use of motility medications or dietary fiber supplements within 2 weeks before study start, taking anti-coagulant drugs, plans to have surgery during the time of the study, a history of alcohol or drug abuse, participation in another clinical trial during 3 months prior the study entry. Patients were not allowed to take antibiotics during the study. Those who were treated with antibiotics were excluded. Patients were allowed to take spasmolytic drugs on an ad hoc basis and/or low-dose antidepressants up to 25 mg per day of amitriptyline, nortriptyline, or selective serotonin inhibitor.

### 2.3. The Multi-Strain Probiotic Preparation

The probiotic in the form of capsules contained a multi-strain mixture of four *Bifidobacterium*, five *Lactobacillus*, and one *Streptococcus thermophilus* species (Table 1). There were two and a half billion of bacteria in each capsule. The placebo capsules contained maltodextrin comparable to the probiotic mixture in color, texture, and taste. The probiotic preparation and placebo were supplied by MBM Biotix, Warsaw, Poland. The samples were marked on the collective packaging as products C and D. The packaging and samples looked identical and contained the title of the trial, the number of the Bioethical Committee agreement, the batch number, and expiry date. Two batches of each product were produced and had a 2-year shelf life. The products were stored below 6 °C until they were delivered to study doctors, where they were stored at room temperature. The products were refrigerated for no longer than 6 months and were issued to doctors every 2–3 months, as needed.

### 2.4. The Study Protocol

During screening visits, the patients underwent physical examination to establish the presence of clinical inclusion criteria. Out of 102 patients with IBS, 76 met the inclusion criteria for IBS severity (IBS-SSS score > 175), with 17 of them not meeting some of the other inclusion criteria and a further 8 not agreeing to participate in the study (Figure 1). In consequence, 51 patients meeting all inclusion criteria signed the informed consent form. All the participants were instructed on how to assess the symptoms for which they would be asked by telephone interviewers and were trained not to consume foods and dietary supplements containing probiotics. During the first baseline visit, patients after evaluation with the use of a IBS-SSS scale were allocated to either the probiotic or placebo group according to a computer-generated randomization list. The block randomization method was used to ensure a balance in sample size across groups over time [34]. Each block had a size of 2n, where n was the number of patients supplemented with probiotic or placebo (in our study, patients were supplemented with either product C or D). The randomization list was generated assuming that the number *n* = 2, and possible treatment allocations within each block were CCDD, DDCC, CDCD, DCDC, CDDC, and DCCD. The block size was not disclosed to the investigators, and the allocation was blinded to both patients and investigators. Patients were asked to take orally one capsule of the probiotic preparation or placebo 2 times a day over an 8-week period. Patients reported to a physician every 4 weeks in order to receive the probiotic preparation or placebo for the next 4 weeks and to be clinically assessed.

The telephone interviewers called patients 3 times per week and collected the information on IBS symptoms, taking drugs, and the occurrence of any adverse events. The recruiting of telephone interviewers took place among the research staff of the Children’s Memorial Health Institute in Warsaw, and all held a PhD degree.

### 2.5. Endpoint Definitions

The primary outcomes included an assessment of IBS symptom severity and improvement using the IBS-SSS and an evaluation of the improvement or exacerbation of global IBS symptoms using the Global Improvement Scale (IBS-GIS).

IBS-SSS is a 5-question survey about the severity of abdominal pain (IBS-SSS1), the number of days with abdominal pain over the last 10 days (IBS-SSS2), the severity of abdominal distension (IBS-SSS3), dissatisfaction with bowel habit (IBS-SSS4), and interference with the quality of life over the past 10 days (IBS-SSS5) [33]. Patients rated each of these aspects on a visual analogue scale from 0–100 points, scoring a maximum of 500 points. The higher scores indicated severe symptoms and total IBS-SSS potentially ranged from 0 to 500 points. A 30% drop in scores compared with baseline was assumed to be associated with a clinically meaningful improvement.

Additionally, patients assessed their IBS symptoms using the IBS-GIS. IBS-GIS included one question about how they felt about the change in severity of symptoms in the last 7 days compared to how they felt before the intervention [35]. The answers were recorded based on a patient defined 7-point Likert scale. A IBS-GIS score of >4 points indicated an improvement of symptoms, <4 points indicated a worsening, and 4 points indicated no change.

Secondary outcomes included changes in stool consistency evaluated using the Bristol Stool Form scale, the number of bowel movements per day, the severity of pain and flatulence, fecal urgency, the feeling of the incomplete evacuation of stool, and the effect of the intervention on the taking of drugs and the occurrence of adverse events. The data were collected by telephone interviewers 3 times a week before the interventions and during the intervention. IBS symptoms, except for the sensation of incomplete bowel movements were assessed using a patient-defined 5-point Likert scale, as was described in our previous paper [36]. Shortly, 0 indicated the absence of symptoms and scores 1–4 depended on the severity of symptoms; the higher the score, the more severe the symptoms. The sensation of incomplete bowel movements was evaluated with a 2-point score (0—no such sensation, 1—the presence of an incomplete bowel movement). Adverse events and taking of drugs were evaluated as either 0—no or 1—yes.

### 2.6. Statistics

The Stata Program version 12.1 was used for statistical analysis. The differences between the probiotic and placebo groups in terms of the sex and number of patients with an improvement, adverse symptoms, or those taking drugs were evaluated with the use of Fisher’s exact test. The intergroup and intragroup differences in age, physical development parameters, and data on IBS symptoms across subsequent visits or weekly telephone interviews were evaluated with two-sided paired or unpaired t-tests or RM ANOVA after checking for the equality of variances and normality using a Shapiro–Wilk test. If the data were not normally distributed, two-sample Wilcoxon paired or unpaired signed-rank tests were used. The threshold of significance for all analyses was set to α = 0.05.

## 3. Results

### 3.1. Patients

A total of 51 patients were randomized to receive multi-strain probiotic preparation or placebo (Figure 1). After a 8-week intervention period, three patients from the placebo group were excluded because of antibiotic therapy (*n* = 2) and no telephone contact with an interviewer (*n* = 1). Nobody was excluded from the probiotic group. Thus, a total of 48 patients (25 supplemented the probiotic preparation and 23 receiving placebo) completed the study. The characteristics of the patients are shown in Table 2. Females were predominant in both groups, at 68.0% and 60.9% in the probiotic and placebo group, respectively. There were no significant differences between the probiotic and placebo group in terms of patient sex, age, physical development, or IBS severity.

### 3.2. Changes in the IBS-SSS Score

The primary outcomes included changes in IBS symptom severity evaluated with the use of the IBS-SSS. The mean values of the total IBS-SSS score before the intervention were similar in both groups (339.6 ± 76.8 and 325.9 ± 50.7 in probiotic and placebo groups, respectively) (Table 3). The probiotic treatment induced a significantly greater reduction in the total IBS-SSS score (reflecting symptom amelioration) in comparison with the placebo. This beneficial effect was observed after 8 weeks of treatment. The total IBS-SSS score was significantly lower, indicating better symptom improvement in the probiotic group compared with the placebo group, when both the mean score value (173.8 ± 60.9 in the probiotic group and 220.3 ± 55.7 in the placebo group, *p* = 0.008) and the change from baseline (−165.8 ± 78.9 in the probiotic group and −105.6 ± 60.2 in the placebo group, *p* = 0.005) were assessed (Table 3 and Table 4). At the end of the study, 60% of the patients in the probiotic group rated their symptoms as mild compared with 30.4% of those in the placebo group (*p* = 0.048) (Table 3).

Probiotic intervention induced a significant reduction in IBS- SSS scores related to the severity of pain (IBS-SSS1) and the quality of life (IBS-SSS5). Patients in the probiotic group compared with the placebo group reported statistically less pain (score change from baseline −30.0 ± 22.8 in probiotic group and −13.0 ± 19.7 in placebo group, *p* = 0.015) and an improved quality of life (score change from baseline −42.9 ± 26.5 in probiotic group and −20.5 ± 26.3 in placebo group, *p* = 0.016) after 8 weeks of treatment (Table 4). In addition, although patients in both groups experienced a systematic improvement in most IBS symptoms during the 8 weeks of treatment, when assessing the frequency of pain (IBS-SSS2) patients in the placebo group in contrast to patients in the probiotic group reported no improvement after 4 weeks of intervention and a slight reduction in the incidence of pain (not statistically significant) after 8 weeks (Table 4).

A beneficial effect of probiotics on course of IBS, especially the severity of pain, was also observed when the number of patients who reported an improvement assessed with the use of IBS-SSS scale was analyzed (Figure 2). An improvement was considered to be when the score after 4 and 8 weeks of treatment decreased by at least 30% compared with the baseline. After 8 weeks of treatment, the percentage of respondents reporting an improvement in most symptoms, except for the severity of flatulence, was greater in the probiotic group, but statistical significance between the groups was obtained for the total IBS-SSS score (*p* = 0.017) and IBS-SSS1 score related to the severity of pain (*p* = 0.038).

### 3.3. Global IBS Symptoms Assessed with the IBS-GIS

The results indicating a beneficial impact of the probiotic intervention on IBS symptom severity evaluated with the IBS-SSS scale with respect to baseline scores were confirmed with the use of the IBS-GIS. Patients from the probiotic group rated the improvement in symptom severity as significantly greater than placebo-group patients, both after 4 weeks (*p* = 0.04) and 8 weeks (*p* = 0.003) of intervention (Figure 3a). According to the study design, an improvement in symptoms was a rating of >4 points on the IBS-GIS, a rating of 4 meant a lack of improvement, and a rating of <4 indicated a worsening. Consequently, the percentage of patients rating the improvement in their symptoms at >4 points was significantly higher in the probiotic group compared with the placebo group at both treatment time points (Figure 3b).

### 3.4. Secondary Outcomes

The secondary endpoints included the impact of the probiotic intervention on severity of selected symptoms. The data were collected by tele-interviewers. During the 8-week intervention, a significant improvement was observed in both the probiotic and placebo group, but without statistical differences between study groups (data not shown). The mean number of bowel movements per day at baseline of 3.38 and 2.98 decreased systematically to values of 2.19 and 2.01 in the probiotic and placebo groups, respectively. Similarly, a statistically significant improvement in stool consistency assessed using the Bristol Stool Form scale was observed. At baseline, stools were assessed as diarrheal with mean values of 6.4 in the probiotic group and 6.2 in the placebo group. After 8 weeks of intervention, there was a significant improvement and the mean values indicated normal stool consistency, with mean values of 4.34 and 4.68 in the probiotic and placebo groups, respectively. An improvement was also reported in the severity of pain, flatulence, fecal urgency, and the sensation of incomplete bowel movements. There were no differences between study groups in the use of analgesics and spasmolytic drugs, with four patients in the probiotic group and five in the placebo group reporting the use of such drugs (*p* = 0.719).

### 3.5. Safety and Adverse Events

The multi-strain probiotic was well tolerated by patients. There were no significant differences in the occurrence of adverse events between the study groups. At the beginning of intervention, three participants from the probiotic group and two from the placebo group reported headache and nausea (*p* = 1.000). In both groups, the reported symptoms were resolved by week 8.

## 4. Discussion

The aim of the current RDBPC trial was to determine whether the multi-strain probiotic consists of the mixture of five *Lactobacillus* species (*Lactobacillus rhamnosus* LR110, *Lactobacillus paracasei* LPC100, *Lactobacillus acidophilus* LA120, *Lactobacillus casei* LC130, *Lactobacillus plantarum* LP140), four *Bifidobacterium* (*Bifidobacterium breve* BB010, *Bifidobacterium longum* BL020, *Bifidobacterium bifidum* BF030, *Bifidobacterium lactis* BL040) species, and *Streptococcus thermophilus* ST250 would be safe and effective in improving global symptom severity (measured with the use of the IBS-SSS and IBS-GIS) in patients with IBS-D. The results of our study showed the probiotic group to be superior to the placebo group in terms of primary outcomes. This was largely due to the observed benefit in improving the severity of abdominal pain and quality of life. Our results are consistent with the findings that multispecies combinations of probiotic strains have significant effects on IBS symptoms [28,29,37,38]. Recently published systemic review and meta-analyses show that multispecies combinations of probiotic strains are superior to single strain preparations [28,29], but in contrast Liang et al. conclude that single probiotics seem to be better choice as they affect specific symptoms—e.g., bloating [25].

Positive effects of multi-strain probiotics in IBS patients were also reported by other researchers. Ishaque et al. also assessed the effectiveness of a probiotic mixture of 14 different species (*Bacillus subtilis*, four *Bifidobacterium* species, seven *Lactobacillus* species, *Lactococcus lactis* and *Streptococcus thermophilus*) in moderate-to-severe IBS-D [37]. They observed a significant decrease in the total IBS-SSS score in the probiotic group, which—like in our study—was mainly due to a highly significant reduction in abdominal pain levels and an improved quality of life. This led to a significant increase in the number of patients with mild IBS (<175 points in the IBS-SSS scale) at the end of study. Our study also showed a decline in the number of patients with severe and moderate types of IBS, and, consequently, the percentage of patients with mild symptoms increased significantly in the probiotic group (reaching 60%) compared with that figure in the placebo group (30.4%). Ishaque et al. reported that 52.5% of patients in the probiotic group rated their symptoms as mild compared with 39.1% of those in the placebo group. In contrast to these results, Sisson et al. [38], who analyzed the effects of the multi-strain probiotic containing three *Lactobacillus* species (*Lactobacillus rhamnosus, Lactobacillus plantarum, Lactobacillus acidophilus*) and *Enterococcus faecium* in IBS patients, reported no difference in the number of patients achieving symptom relief (mild or no symptoms) at the end of the treatment period. However, they found that the evaluated multi-strain probiotic intervention significantly decreased the total IBS-SSS score and the IBS-SSS1 and IBS-SSS4 scores related to the severity of pain and bowel habit satisfaction, but they did not observe the benefit of improving the quality of life [38].

Thus, multi-strain probiotic preparations (containing more than three species of bacteria) have a positive effect on the clinical course of IBS, reducing the severity of symptoms, including abdominal pain, which is of great importance for the patient’s quality of life. However, not all multi-strain probiotics have beneficial effects. Hod et al. presented that a probiotic preparation containing 11 different species, including *Lactobacillus rhamnosus, Lactobacillus casei*, *Lactobacillus paracasei*, *Lactobacillus plantarum*, *Lactobacillus acidophilus*, *Lactobacillus bulgaricus*, *Lactococcus lactis*, *Bifidobacterium bifidum*, *Bifidobacterium longum*, *Bifidobacterium breve*, *Bifidobacterium infantis*, and *Streptococcus thermophilus*, did not demonstrate the superiority of probiotics over placebo after 8 weeks of treatment in women with IBS-D [39]. The above negative result might be caused by a too-high dose of the applied probiotics. The patients enrolled in the study were supplemented daily for 8 weeks with a probiotic at a dose of 50 billion CFU per day. Liang et al. analyzed the effect of probiotic doses in patients with IBS in their systematic review [28]. The authors divided the interventions with multiple probiotic preparation into high-dose (probiotic intake >10^10^ CFU/day) and low-dose (probiotic intake = 10^9^~10^10^ CFU/day) and showed that only a low-dose multi-strain probiotic containing *Lactobacillus* and *Bifidobacterium* species is associated with the improvement of global IBS symptoms. This phenomenon can be explained by the fact that a high number of active probiotic bacteria can induce over-fermentation of carbohydrates with gas production, and discomfort in bowel habits [28].

Our patients received 5 × 10^9^ CFU per day and tolerated this dose of probiotics well. Occasional adverse events including headache and flatulence were reported, but their incidence did not differ between the probiotic group and the placebo group. Taking into account the fact that both headache and flatulence are symptoms that occur in people with IBS, it cannot be ruled out that the reported side effects may not result from taking a probiotic preparation (the more so as they appeared with a similar frequency in placebo group), but constitute the clinical picture of the disease. Unfortunately, before entering the study, no questionnaire was conducted in which the symptoms of IBS in the studied groups were analyzed in detail. The headache may be related to a disturbance in the functioning of the microbiota−gut−brain axis [40].

The effectiveness of probiotics in IBS depends on their composition, dose, and duration of administration. The importance of probiotic composition can be observed by comparing the results of the current study with those of our recently published RDBPC trial in a group of IBS patients [36]. Both studies were conducted according to the same protocol and were performed by the same research teams, but in the study published in 2020 IBS-D patients were treated with a multi-strain synbiotic preparation containing three *Bifidobacterium* species (*Bifidobacterium lactis, Bifidobacterium longum, Bifidobacterium bifidum*), two *Lactobacillus* species (*Lactobacillus rhamnosus, Lactobacillus acidophilus*), and short chain fructooligosaccharides as a prebiotic component. Although the administration of a synbiotic significantly decreased the total IBS-SSS score, it mainly improved the severity of flatulence but had no effect on the perception of pain. A comparison of the results of these two studies may indicate that the probiotic/synbiotic preparation should be matched to the dominant symptoms.

Nevertheless, the length of therapy is also an important aspect. Shavakhi et al., who supplemented IBS patients with a multi-strain probiotic preparation containing four *Lactobacillus* species (*Lactobacillus casei, Lactobacillus rhamnosus, Lactobacillus acidophilus,* and *Lactobacillus bulgaricus*), two *Bifidobacterium* species (*Bifidobacterium breve* and *Bifidobacterium longum)*, and *Streptococcus thermophilus*, observed no beneficial effects on either IBS symptoms, including pain and flatulence, or quality of life in the probiotic group compared with the placebo group [41]. However, this could be due to the too-short intervention period. In this study, the probiotic was administered for only 2 weeks. We observed a statistically significant decrease in the IBS-SSS score in both groups after 4 weeks of treatment, but only after 8 weeks of treatment there was a statistically significant difference between the probiotic and the placebo groups. This observation indicates the need to administer probiotics for more than 4 weeks. When we assessed the global improvement of symptoms on the IBS-GIS, statistically significant differences in favor of the probiotic group were observed as early as after 4 weeks. However, it should be emphasized that the assessment in the IBS-GIS contains the answer to only one question, which may affect the over-interpretation of the obtained results. Thus, it cannot be ruled out that 4 weeks may be the lower limit of a positive response to a multi-strain probiotic. This was confirmed by Ishaque et al. who reported a beneficial effect of multi-strain probiotics assessed with the use of IBS-SSS scale as early as at one month of intervention [37]. The lack of statistically significant differences in the IBS-SSS score between our study groups after 4 weeks of intervention may be due to the very strong placebo effect observed in the study. We associate this effect with relatively frequent and constant contact with telephone interviewers (the same person throughout the entire study), which was often awaited, especially by older patients (such an account was reported by both the patients and the interviewers). It seems that the placebo effect is also responsible for the lack of significant differences between the study groups in the secondary endpoints, especially in the frequency of bowel movements and stool consistency, which are so important for the IBS-D subtype. Both parameters significantly improved during the 8-week observation, but their normalization concerned both the probiotic group and the placebo group.

There is also the question whether 8 weeks of treatment with a multi-strain probiotic preparation is enough to maintain a lasting beneficial effect. If so, how long does it last? However, in our study we did not plan a follow-up after the intervention was completed. Thus, we cannot clearly answer these questions. Ishaque et al. showed that 16-week treatment with a multi-strain probiotic preparation helps maintain a positive effect for at least one month [37]. Sisson et al. [38] showed that the benefits of a 12-week multi-strain treatment lasted for the next 4 weeks.

### Limitations and Strengths of the Study

While our trial was RDBPC, which is its strong point, it also has a number of limitations that need to be outlined. Firstly, the number of patients included in this study was small (51 patients, 48 of whom completed the study). Thus, the statistical power to determine statistically significant differences between the probiotic and placebo groups could be limited. Secondly, the patients included in the study were diagnosed according to the Rome III criteria, which were replaced in 2016 with the current Rome IV criteria [4]. It should be assumed that some of the patients included in the study would not meet the Rome IV criteria, but our study included patients with moderate and severe IBS, which minimizes this eventuality [4]. Other protocol-related limitations relate to the selection of secondary endpoints, especially the assessment of the intervention’s impact on stool consistency and the frequency of bowel movements. Due to the inclusion of IBS-D patients in the study, the improvement of these parameters is important, as it may affect the quality of life. It seems that the assessment of both stool consistency and the frequency of bowel movements should be carried out not only by telephone interviewers, but also by doctors after 4 and 8 months of intervention. The duration of the intervention is also a limitation of this trial. The study treatment was limited to an 8-week period, and there was no follow-up. Consequently, no conclusions regarding a lasting effect of the response can be made. Another limitation of this study was the assessment of the quality of life with the IBS-SSS scale. It seems reasonable to extend the assessment of the impact of probiotics on the quality of life (QOL) using the IBS-QOL-34 questionnaire [42].

The strengths of this RDPC trial include a relatively homogeneous group of patients with a defined subtype and severity of IBS and the systematic contact between a tele-interviewer (the same one) and patient, which ensured a relatively restrictive patient monitoring with a particular emphasis on treatment adherence, adverse events, and taking other medication, including antibiotics. Another strength of the study is the assessment of the probiotic effectiveness using two scales (IBS-SSS and IBS-GIS), which allowed for the analysis of the severity of both the course of IBS and specific symptoms and the improvement of the patient’s condition. In addition, the reasons for study withdrawal and the small number of patients who withdrew also should be considered as strengths of our study.

## 5. Conclusions

The results of this RDBPC trial, in which adult IBS-D patients were involved, indicate that a preparation consisting of *Lactobacillus*, *Bifidobacterium*, and *Streptococcus thermophilus* probiotic strains and administrated at a dose of 5 billion CFU daily for 8 weeks has superior effects in comparison with a placebo. The probiotic was well tolerated, safe, and induced an improvement in global IBS symptoms (evaluated with the IBS-SSS and IBS-GIS scales) as well as specific symptoms related to the severity of pain and the quality of life. Thus, the results show that the multi-strain probiotic preparation offers benefits for adult IBS-D patients. Further studies with longer intervention and follow-up periods will allow us to test whether an 8-week supplementation is sufficient to maintain the achieved beneficial effects.

## Figures and Tables

**Figure 1 nutrients-13-00756-f001:**
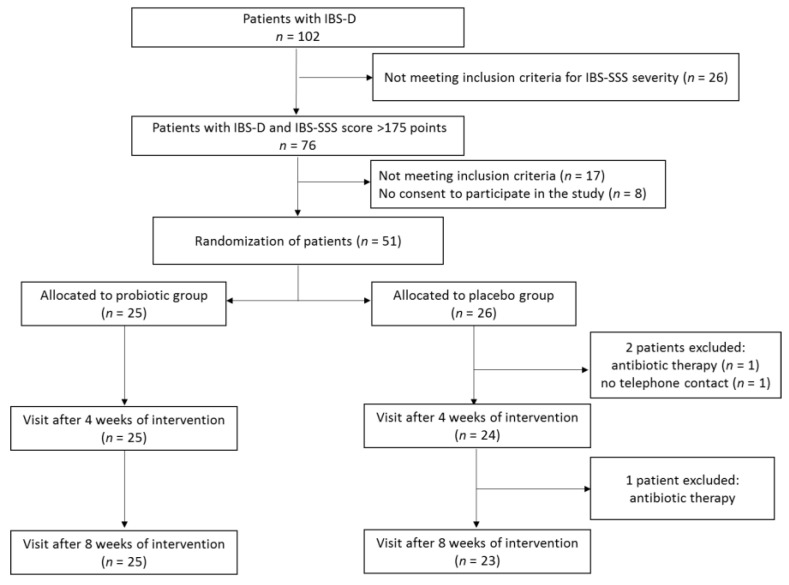
Study protocol flowchart.

**Figure 2 nutrients-13-00756-f002:**
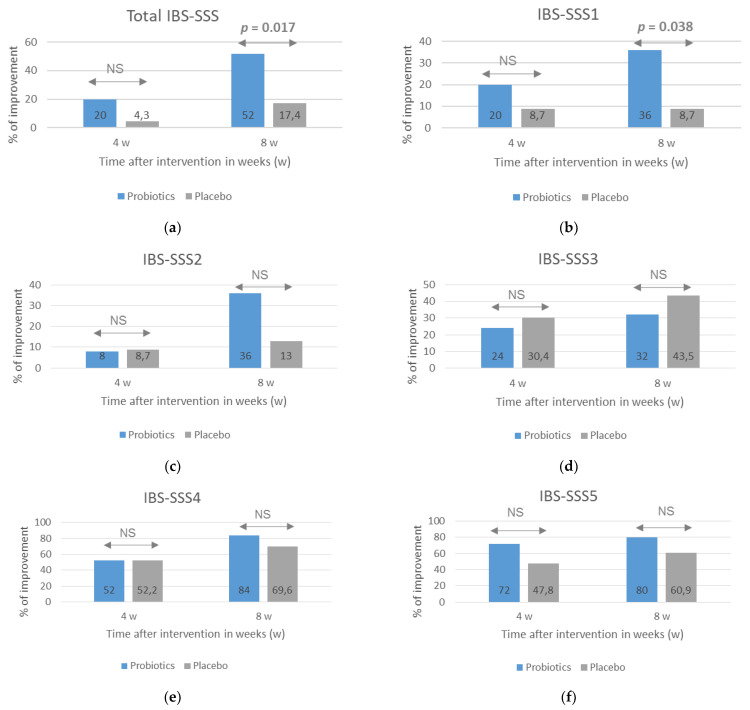
The percentage of patients with an improvement in IBS symptoms evaluated with the use of the IBS-SSS after probiotic treatment lasting 4 and 8 weeks. The total IBS-SSS score (**a**) and domain-specific scores related to the severity of pain (**b**), frequency of pain (**c**), dissatisfaction with bowel habit (**d**), severity of flatulence (**e**), and quality of life were analyzed (**f**). An improvement was defined as at least a 30% decrease in scores compared with baseline scores. NS = no significance (*p* > 0.05). Statistical analysis performed with the use of Fisher’s exact test.

**Figure 3 nutrients-13-00756-f003:**
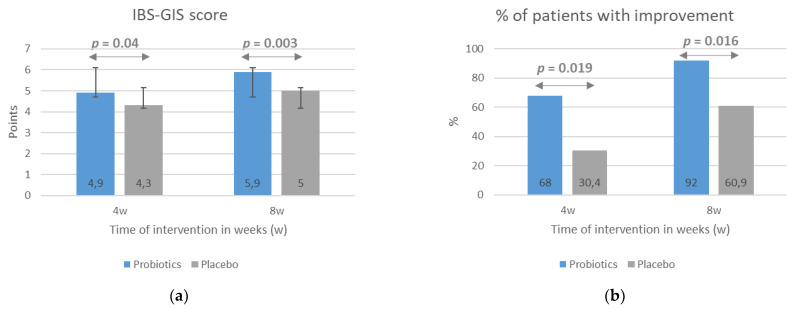
The effect of probiotic intervention on global IBS symptoms evaluated with the use the IBS-GIS. The IBS-GIS score (**a**) and an improvement in global IBS symptoms (**b**) were evaluated after 4 and 8 weeks of treatment. The results are presented as means ± SD of the IBS-GIS score (**a**) or the percentage of patients with an improvement in IBS symptoms (**b**). IBS-GIS scores >4 points indicated an improvement.

**Table 1 nutrients-13-00756-t001:** Specification of strains (NordBiotic™) used in the evaluated probiotic preparation.

Strain	CFU Per Capsule	Strain Number
*Bifidobacterium breve*	2.08 × 10^8^	BB010
*Bifidobacterium longum*	1.39 × 10^8^	BL020
*Bifidobacterium bifidum*	1.39 × 10^8^	BF030
*Bifidobacterium lactis*	4.17 × 10^8^	BL040
*Lactobacillus rhamnosus*	5.56 × 10^8^	LR110
*Lactobacillus paracasei*	2.08 × 10^8^	LPC100
*Lactobacillus acidophilus*	2.08 × 10^8^	LA120
*Lactobacillus casei*	2.08 × 10^8^	LC130
*Lactobacillus plantarum*	2.08 × 10^8^	LP140
*Streptococcus thermophilus*	2.08 × 10^8^	ST250
Total CFU	2.50 × 10^9^	

All the strains are deposited in the patent deposit of the Leibniz Institute DSMZ—German Collection of Microorganisms and Cell Cultures. CFU = colony forming units.

**Table 2 nutrients-13-00756-t002:** Patient characteristics.

	Probiotic Group (*n* = 25) N (%) or Mean ± SD	Placebo Group (*n* = 23) N (%) or Mean ± SD
Sex		
Female	17 (68.0%)	14 (60.9%)
Male	8 (32.0%)	9 (39.1%)
Age (years)	45.5 ± 11.1	40.7 ± 14.4
Growth (cm)	170.4 ± 9.7	168.4 ± 11.0
Body weight (kg)	77.3 ± 15.2	71.1 ± 17.0
BMI	26.5 ± 4.7	25.2 ± 4.2
IBS severity *		
Moderate	4 (16.0%)	7 (30.4%)
Severe	21 (84.0%)	16 (69.6%)
Total IBS-SSS score	339.6 ± 76.8	325.9 ± 50.7

* Irritable bowel syndrome (IBS) severity was evaluated with the use of the IBS Severity Scoring System (IBS-SSS). There were no significant differences between patients receiving a probiotic preparation and placebo. BMI = Body Mass Index.

**Table 3 nutrients-13-00756-t003:** Changes in IBS severity assessed with the use of the IBS-SSS.

Study Time Point/IBS Severity/Total IBS-SSS Score	Probiotic Group (*n* = 25)	Placebo Group (*n* = 23)	*p*-Values
Baseline			
Severe	4 (16.0%)	7 (30.4%)	NS
Moderate	21 (84.0%)	16 (69.6%)	NS
Mild	0	0	
Total IBS-SSS	339.6 ± 76.8	325.9 ± 50.7	NS
4 weeks			
Severe	4 (16.0%)	6 (26.1%)	NS
Moderate	15 (60.0%)	15 (65.2%)	NS
Mild	6 (24.0%)	2 (8.7%)	NS
Total IBS-SSS	235.4 ± 77.9	250.4 ± 60.9	NS
8 weeks			
Severe	0	2 (8.7%)	NS
Moderate	10 (40.0%)	14 (60.9%)	NS
Mild	15 (60.0%)	7 (30.4%)	0.048
Total IBS-SSS	173.8 ± 60.9	220.3 ± 55.7	0.008

The results are shown as the numbers and proportions (in parentheses) of patients with different levels of IBS severity. Total IBS-SSS scores are presented as means ± SD. Statistical analyses were performed with Fisher’s exact test and RM Anova. *p* values < 0.05 represent statistically significant differences. NS = not significant.

**Table 4 nutrients-13-00756-t004:** The impact of probiotic intervention on changes in IBS-SSS scores.

Groups	Baseline	After 4-Week Intervention				After 8-Week Intervention			
	Mean ± SD	Mean ± SD	Change from Baseline	*p*-Value within-Group	*p*-Value Comparison with Placebo	Mean ± SD	Change from Baseline	*p*-Value within-Group	*p*-Value Comparison with Placebo
**Total IBS-SSS**									
Probiotic	339.6 ± 76.8	235.4 ± 77.9	−104.2 ± 69.1	<0.00001	0.159	173.8 ± 60.9*	−165.8 ± 78.9	<0.00001	0.005
Placebo	325.9 ± 50.7	250.4 ± 60.9	−75.5 ± 69.5	<0.00001		220.3 ± 55.7	−105.6 ± 60.2	<0.00001	
**IBS-SSS 1 (the severity of pain)**									
Probiotic	63.0 ± 21.8	42.0 ± 18.7	−21.0 ± 21.3	<0.00001	0.136	33.0 ± 13.9	−30.0 ± 22.8	<0.00001	0.015
Placebo	55.4 ± 19.9	44.6 ± 16.8	−10.9 ± 19.7	0.027		42.4 ± 17.6	−13.0 ± 19.7	0.009	
**IBS-SSS 2 (the frequency of pain)**									
Probiotic	43.0 ± 22.3	29.0 ± 22.4	−14.0 ± 27.1	0.006	0.211	18.0 ± 19.8	−25.0 ± 27.9	0.0003	0.105
Placebo	33.7 ± 20.8	33.7 ± 26.6	−1.1 ± 32.4	0.699		22.8 ± 21.2	−10.9 ± 25.9	0.065	
**IBS-SSS 3 (the severity of flatulence)**									
Probiotic	58.0 ± 31.2	40.0 ± 25.0	−18.0 ± 30.2	0.01	0.165	29.0 ± 20.0	−29.0 ± 22.4	<0.00001	0.594
Placebo	66.3 ± 26.8	38.0 ± 24.8	−28.3 ± 29.5	0.0004		35.9 ± 19.7	−30.4 ± 21.3	<0.00001	
**IBS-SSS 4 (dissatisfaction with bowel habit)**									
Probiotic	86.4±17.0	64.8 ± 18.0	−21.6 ± 23.5	0.0002	0.933	47.5 ± 19.2	−38,88 ± 23.0	<0.00001	0.526
Placebo	88.2 ± 16.5	69.0 ± 17.3	−19.1 ± 24.6	0.003		57.4 ± 14.8	−30.8 ± 24.6	<0.00001	
**IBS-SSS 5 (quality of life)**									
Probiotic	89.2 ± 18.8	59.6 ± 23.6	−29.6 ± 22.3	<0.00001	0.090	46.3 ± 21.5 *	−42.9 ± 26.5	<0.00001	0.016
Placebo	82.3 ± 20.0	66.1 ± 17.5	−16.2 ± 22.3	0.002		61.8 ± 18.3	−20.5 ± 26.3	0.002	

IBS symptom severity was evaluated with the use of the IBS-SSS at baseline and after 4 and 8 weeks of treatment. A score reduction corresponded to symptom amelioration. The results are shown as a mean ± standard deviation (SD). Presented *p*-values are for changes in IBS-SSS scores. *p* < 0.05 represents statistically significant differences. * statistically significant differences in means between the probiotic and placebo groups.

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
