# Peer review of "The Effectiveness and Safety of Multi-Strain Probiotic Preparation in Patients with Diarrhea-Predominant Irritable Bowel Syndrome: A Randomized Controlled Study"

_nutrients, 2021, doi:10.3390/nu13030756_

Round 1

Reviewer 1 Report

This  randomized, double-blind, placebo-controlled trial assessed  the effectiveness of multi-strain probiotic mixture in adults with diarrhea-predominant irritable bowel syndrome (IBS-D).  Enrolling 51 patients, authors found that a 8-weeks course with a probiotic mixture was better than placebo in improving significantly the IBS Severity Scoring System (IBS-19 SSS) and the Global Improvement Scale (IBS-GIS), as well as the specific scores related to the severity of pain and the quality of life.

This study is of interest and worthy of acceptance. However, some points have to be addressed:

  1. For how many weeks were the antibiotics excluded?
  2. Did the authors excluded also parasitic or bacterial intestinal infestation/infection?
  3. The allocation of the patients according to a computer-generated blocked list is quite unclear. Please explain it better.
  4. How the authors explain the occurrence of headache in the study group?
  5. The analysis of the secondary points did not find any significant effect of this probiotic mixture on the bowel movements per day. Since this study enrolled IBS-D patients, this is not a simple secondary point, but it is an important endpoint to reach because it can influence significantly the quality of life of those patients. This endopoint has therefore to be analyzed and discussed more in depth.

Author Response

Answer to the Reviewer 1

All authors would like to thank you for revision of our manuscript and all comments and suggestion which allow us to improve the manuscript. Changes made are marked in red (added in the track changes mode). Below are our answers to all comments:

  1. For how many weeks were the antibiotics excluded?

In the revised manuscript, we accurately described how long patients were not allowed to use antibiotics prior to inclusion in the study (Line 122-124). If the patient had taken antibiotics, e.g. rifaximine, he/she had to stop taking this antibiotic for 12 weeks before he was included in the study (according to the inclusion criteria). If patients had antibiotic therapy during the study, such patient was excluded from the study (Line 129).

  1. Did the authors excluded also parasitic or bacterial intestinal infestation/infection?

Thank you for this comment. Yes, we excluded, and in revised version we added this exclusion (Line 121).

  1. The allocation of the patients according to a computer-generated blocked list is quite unclear. Please explain it better.

The allocation of patients has been described in more details, especially the block randomization methods (with given reference no 34).. In the revised manuscript the description of allocation (lines 158-163) is following:

“The block randomization method was used to ensure a balance in sample size across groups over time [34]. Each block had a size of 2n, where n was the number of patients supplemented with probiotic or placebo (in our study patients were supplemented with either product C or D). The randomization list was generated assuming that the number n = 2, and possible treatment allocations within each block were CCDD, DDCC, CDCD, DCDC, CDDC and DCCD”.

  1. How the authors explain the occurrence of headache in the study group?

Thank you for this comment. Headache which was reported as adverse events, now in revised manuscript is discussed in more detail in Discussion section (line 369-376). We suggests that both reported adverse events (headache and flatulence) We suggest that these may not necessarily be side effects as such symptoms may present as a clinical picture of IBS. We suspect that the headache may be related to the disturbance of the microbiota-gut-brain axis. In the revised version of the manuscript, this is described as follows:

“Taking into account the fact that both headache and flatulence, are symptoms that occur in people with IBS, it cannot be ruled out that the reported side effects may not result from taking a probiotic preparation (the more so as they appeared with a similar frequency in placebo group), but constitute the clinical picture of the disease. Unfortunately, before entering the study, no questionnaire was conducted in which the symptoms of IBS in the studied groups were analyzed in detail. The headache may be related to a disturbance in the functioning of the microbiota-gut-brain axis [40].”

  1. The analysis of the secondary points did not find any significant effect of this probiotic mixture on the bowel movements per day. Since this study enrolled IBS-D patients, this is not a simple secondary point, but it is an important endpoint to reach because it can influence significantly the quality of life of those patients. This endpoint has therefore to be analyzed and discussed more in depth.

Thank you very much for this attention. In the revised version of the manuscript, we described the secondary endpoints with particular regard to the stool consistency assessment and the number of stools per day in more detail presenting the mean values of these parameters in study groups (lines 283-290). In addition, we discuss these results in the Discussion section (lines 414-419), where we discuss the lack of differences between the study groups (strong placebo effect, as noted by Reviewer 2). In addition, considering the fact that these symptoms are relevant for IBS-D in the “Limitations and strengths of the study” section - we wrote that including these symptoms in the secondary endpoints is a limitation of the study (lines 436-442).

Added information in the revised version:

Line 283-290:

“The mean number of bowel movements per day at baseline of 3.38 and 2.98, decreased systematically to values of 2.19 and 2.01 in the probiotic and placebo groups, respectively. Similarly, a statistically significant improvement in stool consistency assessed using the Bristol Stool Form scale was observed. At baseline stools were assessed as diarrheal with mean values of 6.4 in the probiotic group and 6.2 in the placebo group. After 8 weeks of intervention there was a significant improvement and the mean values indicated normal stool consistency with mean values 4.34 and 4.68 in the probiotic and placebo groups, respectively”.

Line: 414-419:

“It seems that the placebo effect is also responsible for the lack of significant differences between the study groups in the secondary endpoints, especially in the frequency of bowel movements and stool consistency, which are so important for the IBS-D subtype. Both parameters significantly improved during the 8-week observation, but their normalization concerned both the probiotic group and the placebo group.”

Line 436-442:

“Other protocol-related limitations relate to the selection of secondary endpoints, especially the assessment of the intervention's impact on stool consistency and the frequency of bowel movements. Due to the inclusion of IBS-D patients in the study, the improvement of these parameters are important, ale may affect the quality of life. It seems that the assessment of both stool consistency and the frequency of bowel movements should be carried out not only by telephone interviewers, but also by doctors after 4 and 8 months of intervention.

Reviewer 2 Report

Thank you for this really nice manuscript on probiotics in IBS. I only have minor comments to improve mostly the introduction. The manuscript is well-written clear and provide nice results on the probiotic in the IBS-D subgroup.

Comments:

Introduction:

-The worldwide prevalence according to Rome IV criteria is around 4% see (Gastroenterology. 2021 Jan;160(1):99-114.e3. doi: 10.1053/j.gastro.2020.04.014. Epub 2020 Apr 12. PMID: 32294476)

-IBS-D is more 1/3 of the population but with more severe outcome.

-before to explain the role of the microbiota you should state that IBS is a multifactorial disease and describe the main abnormalities.

-Before to detail the lower abundance of some species you should probably explain what is a dysbiosis.

Methods:

Why have you chosen to use Rome III criteria while Rome IV are available since 2016?

-The definition for IBS-D also excluded associated constipation.

Discussion:

-you should probably add one sentence regarding the absence of superiority between multi-strain probiotics and single strain probiotics.

-please discuss also the placebo effect that is really important during the 1st month of treatment and could also explain the absence of difference at one month!

-one limitation is the use of Rome III criteria instead of rome IV.

Author Response

(The authors gave the same response as above.)

Round 2

Reviewer 1 Report

The authors replied successfully to all concerns raised, and the manuscript may be accepted for publication.  

As minor point, please change “Roman” into “Roma” when describing the IBS criteria (it is a noun,  not an adjective).